# Edge-exchangeable graphs and sparsity

**Diana Cai**
Dept. of Statistics, U. Chicago
Chicago, IL 60637
dcai@uchicago.edu

**Trevor Campbell**
CSAIL, MIT
Cambridge, MA 02139
tdjc@mit.edu

**Tamara Broderick**
CSAIL, MIT
Cambridge, MA 02139
tbroderick@csail.mit.edu

## Abstract

Many popular network models rely on the assumption of *(vertex) exchangeability*, in which the distribution of the graph is invariant to relabelings of the vertices. However, the Aldous-Hoover theorem guarantees that these graphs are dense or empty with probability one, whereas many real-world graphs are sparse. We present an alternative notion of exchangeability for random graphs, which we call *edge exchangeability*, in which the distribution of a graph sequence is invariant to the order of the edges. We demonstrate that edge-exchangeable models, unlike models that are traditionally vertex exchangeable, can exhibit sparsity. To do so, we outline a general framework for graph generative models; by contrast to the pioneering work of Caron and Fox [12], models within our framework are stationary across steps of the graph sequence. In particular, our model grows the graph by instantiating more latent atoms of a single random measure as the dataset size increases, rather than adding new atoms to the measure.

## 1 Introduction

In recent years, network data have appeared in a growing number of applications, such as online social networks, biological networks, and networks representing communication patterns. As a result, there is growing interest in developing models for such data and studying their properties. Crucially, individual network data sets also continue to increase in size; we typically assume that the number of vertices is unbounded as time progresses. We say a graph sequence is *dense* if the number of edges grows quadratically in the number of vertices, and a graph sequence is *sparse* if the number of edges grows sub-quadratically as a function of the number of vertices. Sparse graph sequences are more representative of real-world graph behavior. However, many popular network models (see, e.g., Lloyd et al. [19] for an extensive list) share the undesirable scaling property that they yield dense sequences of graphs with probability one. The poor scaling properties of these models can be traced back to a seemingly innocent assumption: that the vertices in the model are *exchangeable*, that is, any finite permutation of the rows and columns of the graph adjacency matrix does not change the distribution of the graph. Under this assumption, the Aldous-Hoover theorem [1, 16] implies that such models generate dense or empty graphs with probability one [20].

This fundamental model misspecification motivates the development of new models that can achieve sparsity. One recent focus has been on models in which an additional parameter is employed to uniformly decrease the probabilities of edges as the network grows (e.g., Bollobás et al. [3], Borgs et al. [4, 5], Wolfe and Olhede [24]). While these models allow sparse graph sequences, the sequences are no longer *projective*. In projective sequences, vertices and edges are added to a graph as a graph sequence progresses—whereas in the models above, there is not generally any strict subgraph relationship between earlier graphs and later graphs in the sequence. Projectivity is natural in streaming modeling. For instance, we may wish to capture new users joining a social network and new connections being made among existing users—or new employees joining a company and new communications between existing employees.

Caron and Fox [12] have pioneered initial work on sparse, projective graph sequences. Instead of the *vertex exchangeability* that yields the Aldous-Hoover theorem, they consider a notion of graph exchangeability based on the idea of independent increments of subordinators [18], explored in depth by Veitch and Roy [22]. However, since this Kallenberg-style exchangeability introduces a new countable infinity of latent vertices at every step in the graph sequence, its generative mechanism seems particularly suited to the non-stationary domain. By contrast, we are here interested in exploring *stationary* models that grow in complexity with the size of the data set. Consider classic Bayesian nonparametric models as the Chinese restaurant process (CRP) and Indian buffet process (IBP); these engender growth by using a single infinite latent collection of parameters to generate a finite but growing set of instantiated parameters. Similarly, we propose a framework that uses a single infinite latent collection of vertices to generate a finite but growing set of vertices that participate in edges and thereby in the network. We believe our framework will be a useful component in more complex, non-stationary graphical models—just as the CRP and IBP are often combined with hidden Markov models or other explicit non-stationary mechanisms. Additionally, Kallenberg exchangeability is intimately tied to continuous-valued labels of the vertices, and here we are interested in providing a characterization of the graph sequence based solely on its topology.

In this work, we introduce a new form of exchangeability, distinct from both vertex exchangeability and Kallenberg exchangeability. In particular, we say that a graph sequence is *edge exchangeable* if the distribution of any graph in the sequence is invariant to the *order* in which edges arrive—rather than the order of the vertices. We will demonstrate that edge exchangeability admits a large family of sparse, projective graph sequences.

In the remainder of the paper, we start by defining dense and sparse graph sequences rigorously. We review vertex exchangeability before introducing our new notion of edge exchangeability in Section 2, which we also contrast with Kallenberg exchangeability in more detail in Section 4. We define a family of models, which we call *graph frequency models*, based on random measures in Section 3. We use these models to show that edge-exchangeable models can yield sparse, projective graph sequences via theoretical analysis in Section 5 and via simulations in Section 6. Along the way, we highlight other benefits of the edge exchangeability and graph frequency model frameworks.

## 2   Exchangeability in graphs: old and new

Let $(G_n)_n := G_1, G_2, \ldots$ be a sequence of graphs, where each graph $G_n = (V_n, E_n)$ consists of a (finite) set of vertices $V_n$ and a (finite) multiset of edges $E_n$. Each edge $e \in E_n$ is a set of two vertices in $V_n$. We assume the sequence is *projective*—or growing—so that $V_n \subseteq V_{n+1}$ and $E_n \subseteq E_{n+1}$. Consider, e.g., a social network with more users joining the network and making new connections with existing users. We say that a graph sequence is *dense* if $|E_n| = \Omega(|V_n|^2)$, i.e., the number of edges is asymptotically lower bounded by $c \cdot |V_n|^2$ for some constant $c$. Conversely, a sequence is *sparse* if $|E_n| = o(|V_n|^2)$, i.e., the number of edges is asymptotically upper bounded by $c \cdot |V_n|^2$ for all constants $c$. In what follows, we consider random graph sequences, and we focus on the case where $|V_n| \to \infty$ almost surely.

### 2.1   Vertex-exchangeable graph sequences

If the number of vertices in the graph sequence grows to infinity, the graphs in the sequence can be thought of as subgraphs of an "infinite" graph with infinitely many vertices and a correspondingly infinite adjacency matrix. Traditionally, exchangeability in random graphs is defined as the invariance of the distribution of any finite submatrix of this adjacency matrix—corresponding to any finite collection of vertices—under finite permutation. Equivalently, we can express this form of exchangeability, which we henceforth call *vertex exchangeability*, by considering a random sequence of graphs $(G_n)_n$ with $V_n = [n]$, where $[n] := \{1, \ldots, n\}$. In this case, only the edge sequence is random. Let $\pi$ be any permutation of the integers $[n]$. If $e = \{v, w\}$, let $\pi(e) := \{\pi(v), \pi(w)\}$. If $E_n = \{e_1, \ldots, e_m\}$, let $\pi(E_n) := \{\pi(e_1), \ldots, \pi(e_m)\}$.

**Definition 2.1.** Consider the random graph sequence $(G_n)_n$, where $G_n$ has vertices $V_n = [n]$ and edges $E_n$. $(G_n)_n$ is (infinitely) *vertex exchangeable* if for every $n \in \mathbb{N}$ and for every permutation $\pi$ of the vertices $[n]$, $G_n \stackrel{\text{d}}{=} \tilde{G}_n$, where $\tilde{G}_n$ has vertices $[n]$ and edges $\pi(E_n)$.

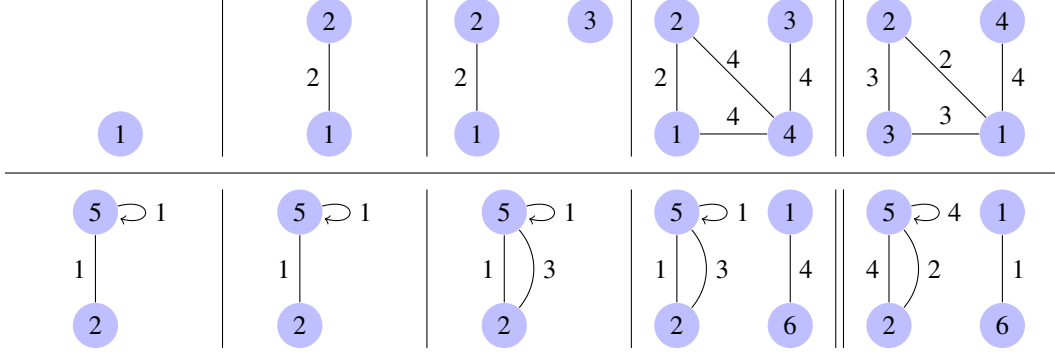

Figure 1: *Upper, left four*: Step-augmented graph sequence from Ex. 2.2. At each step $n$, the step value is always at least the maximum vertex index. *Upper, right two*: Two graphs with the same probability under vertex exchangeability. *Lower, left four*: Step-augmented graph sequence from Ex. 2.3. *Lower, right two*: Two graphs with the same probability under edge exchangeability.

A great many popular models for graphs are vertex exchangeable; see Appendix B and Lloyd et al. [19] for a list. However, it follows from the Aldous-Hoover theorem [1, 16] that any vertex-exchangeable graph is a mixture of sampling procedures from *graphons*. Further, any graph sampled from a graphon is almost surely dense or empty [20]. Thus, vertex-exchangeable random graph models are misspecified models for sparse network datasets, as they generate dense graphs.

## 2.2 Edge-exchangeable graph sequences

Vertex-exchangeable sequences have distributions invariant to the order of vertex arrival. We introduce *edge-exchangeable* graph sequences, which will instead be invariant to the order of edge arrival. As before, we let $G_n = (V_n, E_n)$ be the $n$th graph in the sequence. Here, though, we consider only *active vertices*—that is, vertices that are connected via some edge. That lets us define $V_n$ as a function of $E_n$; namely, $V_n$ is the union of the vertices in $E_n$. Note that a graph that has sub-quadratic growth in the number of edges as a function of the number of active vertices will necessarily have sub-quadratic growth in the number of edges as a function of the number of all vertices, so we obtain strictly stronger results by considering active vertices. In this case, the graph $G_n$ is completely defined by its edge set $E_n$.

As above, we suppose that $E_n \subseteq E_{n+1}$. We can emphasize this projectivity property by augmenting each edge with the step on which it is added to the sequence. Let $E'_n$ be a collection of tuples, in which the first element is the edge and the second element is the step (i.e., index) on which the edge is added: $E'_n = \{(e_1, s_1), \ldots, (e_m, s_m)\}$. We can then define a *step-augmented graph sequence* $(E'_n)_n = (E'_1, E'_2, \ldots)$ as a sequence of step-augmented edge sets. Note that there is a bijection between the step-augmented graph sequence and the original graph sequence.

**Example 2.2.** In the setup for vertex exchangeability, we assumed $V_n = [n]$ and every edge is introduced as soon as both of its vertices are introduced. In this case, the step of any edge in the step-augmented graph is the maximum vertex value. For example, in Figure 1, we have

$$E'_1 = \emptyset, E'_2 = E'_3 = \{(\{1, 2\}, 2)\}, E'_4 = \{(\{1, 2\}, 2), (\{1, 4\}, 4), (\{2, 4\}, 4), (\{3, 4\}, 4)\}.$$

In general step-augmented graphs, though, the step need not equal the max vertex, as we see next. ∎

**Example 2.3.** Suppose we have a graph given by the edge sequence (see Figure 1):

$$E_1 = E_2 = \{\{2, 5\}, \{5, 5\}\}, E_3 = E_2 \cup \{\{2, 5\}\}, E_4 = E_3 \cup \{\{1, 6\}\}.$$

The step-augmented graph $E'_4$ is $\{(\{2, 5\}, 1), (\{5, 5\}, 1), (\{2, 5\}, 3), (\{1, 6\}, 4)\}$. ∎

Roughly, a random graph sequence is edge exchangeable if its distribution is invariant to finite permutations of the steps. Let $\pi$ be a permutation of the integers $[n]$. For a step-augmented edge set $E'_n = \{(e_1, s_1), \ldots, (e_m, s_m)\}$, let $\pi(E'_n) = \{(e_1, \pi(s_1)), \ldots, (e_m, \pi(s_m))\}$.

**Definition 2.4.** Consider the random graph sequence $(G_n)_n$, where $G_n$ has step-augmented edges $E'_n$ and $V_n$ are the active vertices of $E_n$. $(G_n)_n$ is (infinitely) *edge exchangeable* if for every $n \in \mathbb{N}$

and for every permutation $\pi$ of the steps $[n]$, $G_n \stackrel{\text{d}}{=} \tilde{G}_n$, where $\tilde{G}_n$ has step-augmented edges $\pi(E_n')$ and associated active vertices.

See Figure 1 for visualizations of both vertex exchangeability and edge exchangeability. It remains to show that there are non-trivial models that are edge exchangeable (Section 3) and that edge-exchangeable models admit sparse graphs (Section 5).

## 3   Graph frequency models

We next demonstrate that a wide class of models, which we call *graph frequency models*, exhibit edge exchangeability. Consider a latent infinity of vertices indexed by the positive integers $\mathbb{N} = \{1, 2, \ldots\}$, along with an infinity of edge labels $(\theta_{\{i,j\}})$, each in a set $\Theta$, and positive edge rates (or frequencies) $(w_{\{i,j\}})$ in $\mathbb{R}_+$. We allow both the $(\theta_{\{i,j\}})$ and $(w_{\{i,j\}})$ to be random, though this is not mandatory. For instance, we might choose $\theta_{\{i,j\}} = (i, j)$ for $i \leq j$, and $\Theta = \mathbb{R}^2$. Alternatively, the $\theta_{\{i,j\}}$ could be drawn iid from a continuous distribution such as $\text{Unif}[0, 1]$. For any choice of $(\theta_{\{i,j\}})$ and $(w_{\{i,j\}})$,

$$W := \sum_{\{i,j\}:i,j\in\mathbb{N}} w_{\{i,j\}} \delta_{\theta_{\{i,j\}}} \tag{1}$$

is a *measure* on $\Theta$. Moreover, it is a discrete measure since it is always atomic. If either $(\theta_{\{i,j\}})$ or $(w_{\{i,j\}})$ (or both) are random, $W$ is a *discrete random measure* on $\Theta$ since it is a random, discrete-measure-valued element. Given the edge rates (or frequencies) $(w_{\{i,j\}})$ in $W$, we next show some natural ways to construct edge-exchangeable graphs.

**Single edge per step.**   If the rates $(w_{\{i,j\}})$ are normalized such that $\sum_{\{i,j\}:i,j\in\mathbb{N}} w_{\{i,j\}} = 1$, then $(w_{\{i,j\}})$ is a distribution over all possible vertex pairs. In other words, $W$ is a probability measure. We can form an edge-exchangeable graph sequence by first drawing values for $(w_{\{i,j\}})$ and $(\theta_{\{i,j\}})$—and setting $E_0 = \emptyset$. We recursively set $E_{n+1} = E_n \cup \{e\}$, where $e$ is an edge $\{i, j\}$ chosen from the distribution $(w_{\{i,j\}})$. This construction introduces a single edge in the graph each step, although it may be a duplicate of an edge that already exists. Therefore, this technique generates multigraphs one edge at a time. Since the edge every step is drawn conditionally iid given $W$, we have an edge-exchangeable graph.

**Multiple edges per step.**   Alternatively, the rates $(w_{\{i,j\}})$ may not be normalized. Then $W$ may not be a probability measure. Let $f(m|w)$ be a distribution over non-negative integers $m$ given some rate $w \in \mathbb{R}_+$. We again initialize our sequence by drawing $(w_{\{i,j\}})$ and $(\theta_{\{i,j\}})$ and setting $E_0 = \emptyset$. In this case, recursively, on the $n$th step, start by setting $F = \emptyset$. For every possible edge $e = \{i, j\}$, we draw the multiplicity of the edge $e$ in this step as $m_e \stackrel{\text{ind}}{\sim} f(\cdot|w_e)$ and add $m_e$ copies of edge $e$ to $F$. Finally, $E_{n+1} = E_n \cup F$. This technique potentially introduces multiple edges in each step, in which edges themselves may have multiplicity greater than one and may be duplicates of edges that already exist in the graph. Therefore, this technique generates multigraphs, multiple edges at a time. If we restrict $f$ and $W$ such that finitely many edges are added on every step almost surely, we have an edge-exchangeable graph, as the edges in each step are drawn conditionally iid given $W$.

Given a sequence of edge sets $E_0, E_1, \ldots$ constructed via either of the above methods, we can form a binary graph sequence $\bar{E}_0, \bar{E}_1, \ldots$ by setting $\bar{E}_i$ to have the same edges as $E_i$ except with multiplicity 1. Although this binary graph is not itself edge exchangeable, it inherits many of the properties (such as sparsity, as shown in Section 5) of the underlying edge-exchangeable multigraph.

The choice of the distribution on the measure $W$ has a strong influence on the properties of the resulting edge-exchangeable graph sampled via one of the above methods. For example, one choice is to set $w_{\{i,j\}} = w_i w_j$, where the $(w_i)_i$ are a countable infinity of random values generated according to a *Poisson point process* (PPP). We say that $(w_i)_i$ is distributed according to a Poisson point process parameterized by rate measure $\nu$, $(w_i)_i \sim \text{PPP}(\nu)$, if (a) $\#\{i : w_i \in A\} \sim \text{Poisson}(\nu(A))$ for any set $A$ with finite measure $\nu(A)$ and (b) $\#\{i : w_i \in A_j\}$ are independent random variables across any finite collection of disjoint sets $(A_j)_{j=1}^J$. In Section 5 we examine a particular example of this graph frequency model, and demonstrate that sparsity is possible in edge-exchangeable graphs.

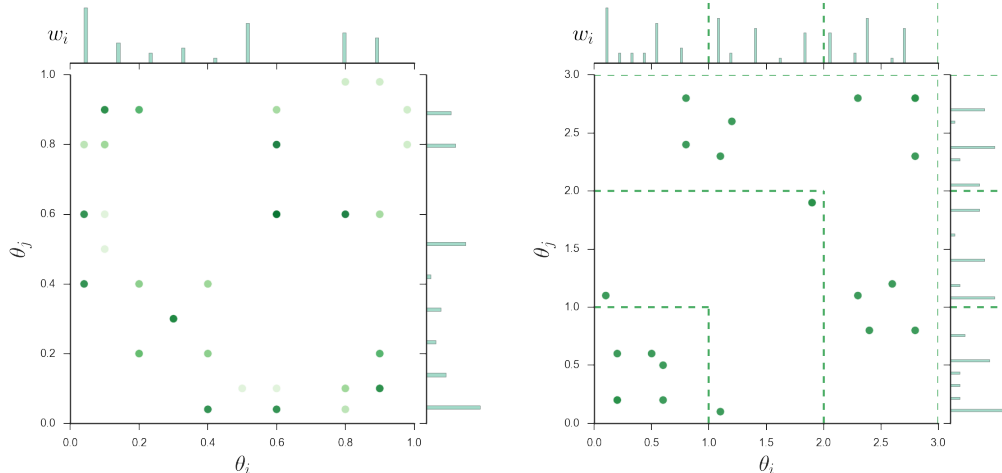

(a) Graph frequency model (fixed $y$, $n$ steps)     (b) Caron–Fox, PPP on $[0, y] \times [0, y]$ (1 step, $y$ grows)

Figure 2: A comparison of a graph frequency model (Section 3 and Equation (2)) and the generative model of Caron and Fox [12]. Any interval $[0, y]$ contains a countably infinite number of atoms with a nonzero weight in the random measure; a draw from the random measure is plotted at the top (and repeated on the right side). Each atom corresponds to a latent vertex. Each point $(\theta_i, \theta_j)$ corresponds to a latent edge. Darker point colors on the left occur for greater edge multiplicities. On the *left*, more latent edges are instantiated as more steps $n$ are taken. On the *right*, the edges within $[0, y]^2$ are fixed, but more edges are instantiated as $y$ grows.

## 4 Related work and connection to nonparametric Bayes

Given a unique label $\theta_i$ for each vertex $i \in \mathbb{N}$, and denoting $g_{ij} = g_{ji}$ to be the number of undirected edges between vertices $i$ and $j$, the graph itself can be represented as the discrete random measure $G = \sum_{i,j} g_{ij} \delta_{(\theta_i, \theta_j)}$ on $\mathbb{R}^2_+$. A different notion of exchangeability for graphs than the ones in Section 2 can be phrased for such atomic random measures: a point process $G$ on $\mathbb{R}^2_+$ is (jointly) exchangeable if, for all finite permutations $\pi$ of $\mathbb{N}$ and all $h > 0$,

$$G(A_i \times A_j) \stackrel{\mathrm{d}}{=} G(A_{\pi(i)} \times A_{\pi(j)}), \text{ for } (i,j) \in \mathbb{N}^2, \qquad \text{where } A_i := [h \cdot (i-1), h \cdot i].$$

This form of exchangeability, which we refer to as *Kallenberg exchangeability*, can intuitively be viewed as invariance of the graph distribution to relabeling of the vertices, which are now embedded in $\mathbb{R}^2_+$. As such it is analogous to vertex exchangeability, but for discrete random measures [12, Sec. 4.1]. Exchangeability for random measures was introduced by Aldous [2], and a representation theorem was given by Kallenberg [17, 18, Ch. 9]. The use of Kallenberg exchangeability for modeling graphs was first proposed by Caron and Fox [12], and then characterized in greater generality by Veitch and Roy [22] and Borgs et al. [6]. Edge exchangeability is distinct from Kallenberg exchangeability, as shown by the following example.

**Example 4.1** (Edge exchangeable but not Kallenberg exchangeable). Consider the graph frequency model developed in Section 3, with $w_{\{i,j\}} = (ij)^{-2}$ and $\theta_{\{i,j\}} = \{i, j\}$. Since the edges at each step are drawn iid given $W$, the graph sequence is edge exchangeable. However, the corresponding graph measure $G = \sum_{i,j} n_{ij} \delta_{(i,j)}$ (where $n_{ij} = n_{ji} \sim \mathrm{Binom}(N, (ij)^{-2})$) is not Kallenberg exchangeable, since the probability of generating edge $\{i, j\}$ is directly related to the positions $(i, j)$ and $(j, i)$ in $\mathbb{R}^2_+$ of the corresponding atoms in $G$ (in particular, the probability is decreasing in $ij$). ∎

Our graph frequency model is reminiscent of the Caron and Fox [12] generative model, but has a number of key differences. At a high level, this earlier model generates a weight measure $W = \sum_{i,j} w_{ij} \delta_{(\theta_i, \theta_j)}$ (Caron and Fox [12] used, in particular, the outer product of a completely random measure), and the graph measure $G$ is constructed by sampling $g_{ij}$ *once* given $w_{ij}$ for each pair $i, j$. To create a finite graph, the graph measure $G$ is restricted to the subset $[0, y] \times [0, y] \subset \mathbb{R}^2_+$ for $0 < y < \infty$; to create a projective growing graph sequence, the value of $y$ is increased. By contrast, in the analogous graph frequency model of the present work, $y$ is fixed, and we grow the network

by *repeatedly* sampling the number of edges $g_{ij}$ between vertices $i$ and $j$ and summing the result. Thus, in the Caron and Fox [12] model, a latent infinity of vertices (only finitely many of which are active) are added to the network each time $y$ increases. In our graph frequency model, there is a single collection of latent vertices, which are all gradually activated by increasing the number of samples that generate edges between the vertices. See Figure 2 for an illustration.

Increasing $n$ in the graph frequency model has the interpretation of both (a) time passing and (b) new individuals joining a network because they have formed a connection that was not previously there. In particular, only latent individuals that will eventually join the network are considered. This behavior is analogous to the well-known behavior of other nonparametric Bayesian models such as, e.g., a Chinese restaurant process (CRP). In this analogy, the Dirichlet process (DP) corresponds to our graph frequency model, and the clusters instantiated by the CRP correspond to the vertices that are active after $n$ steps. In the DP, only latent clusters that will eventually appear in the data are modeled. Since the graph frequency setting is stationary like the DP/CRP, it may be more straightforward to develop approximate Bayesian inference algorithms, e.g., via truncation [11].

Edge exchangeability first appeared in work by Crane and Dempsey [13, 14], Williamson [23], and Broderick and Cai [7, 8], Cai and Broderick [10]. Broderick and Cai [7, 8] established the notion of edge exchangeability used here and provided characterizations via exchangeable partitions and feature allocations, as in Appendix C. Broderick and Cai [7], Cai and Broderick [10] developed a frequency model based on weights $(w_i)_i$ generated from a Poisson process and studied several types of power laws in the model. Crane and Dempsey [13] established a similar notion of edge exchangeability in the context of a larger statistical modeling framework. Crane and Dempsey [13, 14] provided sparsity and power law results for the case where the weights $(w_i)_i$ are generated from a Pitman-Yor process and power law degree distribution simulations. Williamson [23] described a similar notion of edge exchangeability and developed an edge-exchangeable model where the weights $(w_i)_i$ are generated from a Dirichlet process, a mixture model extension, and an efficient Bayesian inference procedure. In work concurrent to the present paper, Crane and Dempsey [15] re-examined edge exchangeability, provided a representation theorem, and studied sparsity and power laws for the same model based on Pitman-Yor weights. By contrast, we here obtain sparsity results across all Poisson point process-based graph frequency models of the form in Equation (2) below, and use a specific three-parameter beta process rate measure only for simulations in Section 6.

## 5  Sparsity in Poisson process graph frequency models

We now demonstrate that, unlike vertex exchangeability, edge exchangeability allows for sparsity in random graph sequences. We develop a class of sparse, edge-exchangeable multigraph sequences via the Poisson point process construction introduced in Section 3, along with their binary restrictions.

**Model.**  Let $\mathcal{W}$ be a Poisson process on $[0, 1]$ with a nonatomic, $\sigma$-finite rate measure $\nu$ satisfying $\nu([0, 1]) = \infty$ and $\int_0^1 w\nu(\mathrm{d}w) < \infty$. These two conditions on $\nu$ guarantee that $\mathcal{W}$ is a countably infinite collection of rates in $[0, 1]$ and that $\sum_{w \in \mathcal{W}} w < \infty$ almost surely. We can use $\mathcal{W}$ to construct the set of rates: $w_{\{i,j\}} = w_i w_j$ if $i \neq j$, and $w_{\{i,i\}} = 0$. The edge labels $\theta_{\{i,j\}}$ are unimportant in characterizing sparsity, and so can be ignored.

To use the multiple-edges-per-step graph frequency model from Section 3, we let $f(\cdot|w)$ be Bernoulli with probability $w$. Since edge $\{i, j\}$ is added in each step with probability $w_i w_j$, its multiplicity $M_{\{i,j\}}$ after $n$ steps has a binomial distribution with parameters $n, w_i w_j$. Note that self-loops are avoided by setting $w_{\{i,i\}} = 0$. Therefore, the graph after $n$ steps is described by:

$$\mathcal{W} \sim \mathrm{PPP}(\nu) \qquad\qquad M_{\{i,j\}} \overset{\mathrm{ind}}{\sim} \mathrm{Binom}(n, w_i w_j) \ \text{ for } i < j \in \mathbb{N}. \tag{2}$$

As mentioned earlier, this generative model yields an edge-exchangeable graph, with edge multiset $E_n$ containing $\{i, j\}$ with multiplicity $M_{\{i,j\}}$, and active vertices $V_n = \{i : \sum_j M_{\{i,j\}} > 0\}$. Although this model generates multigraphs, it can be modified to sample a binary graph $(\bar{V}_n, \bar{E}_n)$ by setting $\bar{V}_n = V_n$ and $\bar{E}_n$ to the set of edges $\{i, j\}$ such that $\{i, j\}$ has multiplicity $\geq 1$ in $E_n$. We can express the number of vertices and edges, in the multi- and binary graphs respectively, as

$$|\bar{V}_n| = |V_n| = \sum_i \mathbb{1}\left(\sum_{j \neq i} M_{\{i,j\}} > 0\right), \quad |E_n| = \frac{1}{2}\sum_{i \neq j} M_{\{i,j\}}, \quad |\bar{E}_n| = \frac{1}{2}\sum_{i \neq j} \mathbb{1}\left(M_{\{i,j\}} > 0\right).$$

**Moments.** Recall that a sequence of graphs is considered *sparse* if $|E_n| = o(|V_n|^2)$. Thus, sparsity in the present setting is an *asymptotic* property of a random graph sequence. Rather than consider the asymptotics of the (dependent) random sequences $|E_n|$ and $|V_n|$ in concert, Lemma 5.1 allows us to consider the asymptotics of their first moments, which are deterministic sequences and can be analyzed separately. We use $\sim$ to denote asymptotic equivalence, i.e., $a_n \sim b_n \iff \lim_{n\to\infty} \frac{a_n}{b_n} = 1$. For details on our asymptotic notation and proofs for this section, see Appendix D.

**Lemma 5.1.** The number of vertices and edges for both the multi- and binary graphs satisfy

$$|\bar{V}_n| = |V_n| \overset{\text{a.s.}}{\sim} \mathbb{E}(|V_n|), \qquad |E_n| \overset{\text{a.s.}}{\sim} \mathbb{E}(|E_n|), \qquad |\bar{E}_n| \overset{\text{a.s.}}{\sim} \mathbb{E}(|\bar{E}_n|), \qquad n \to \infty.$$

Thus, we can examine the asymptotic behavior of the random numbers of edges and vertices by examining the asymptotic behavior of their expectations, which are provided by Lemma 5.2.

**Lemma 5.2.** The expected numbers of vertices and edges for the multi- and binary graphs are

$$\mathbb{E}\left(|\bar{V}_n|\right) = \mathbb{E}\left(|V_n|\right) = \int \left[1 - \exp\left(-\int (1 - (1-wv)^n)\nu(\mathrm{d}v)\right)\right]\nu(\mathrm{d}w),$$

$$\mathbb{E}\left(|E_n|\right) = \frac{n}{2}\iint wv\,\nu(\mathrm{d}w)\nu(\mathrm{d}v), \qquad \mathbb{E}\left(|\bar{E}_n|\right) = \frac{1}{2}\iint \left(1 - (1-wv)^n\right)\nu(\mathrm{d}w)\nu(\mathrm{d}v).$$

**Sparsity.** We are now equipped to characterize the sparsity of this random graph sequence:

**Theorem 5.3.** *Suppose $\nu$ has a regularly varying tail, i.e., there exist $\alpha \in (0,1)$ and $\ell : \mathbb{R}_+ \to \mathbb{R}_+$ s.t.*

$$\int_x^1 \nu(\mathrm{d}w) \sim x^{-\alpha}\ell(x^{-1}), \quad x \to 0 \qquad \text{and} \qquad \forall c > 0, \ \lim_{x\to\infty} \frac{\ell(cx)}{\ell(x)} = 1.$$

*Then as $n \to \infty$,*

$$|V_n| \overset{\text{a.s.}}{=} \Theta(n^\alpha \ell(n)), \qquad |E_n| \overset{\text{a.s.}}{=} \Theta(n), \qquad |\bar{E}_n| \overset{\text{a.s.}}{=} O\left(\ell(n^{1/2})\min\left(n^{\frac{1+\alpha}{2}}, \ell(n)n^{\frac{3\alpha}{2}}\right)\right).$$

Theorem 5.3 implies that the multigraph is sparse when $\alpha \in (1/2, 1)$, and that the restriction to the binary graph is sparse for any $\alpha \in (0,1)$. See Remark D.7 for a discussion. Thus, edge-exchangeable random graph sequences allow for a wide range of sparse and dense behavior.

# 6 Simulations

In this section, we explore the behavior of graphs generated by the model from Section 5 via simulation, with the primary goal of empirically demonstrating that the model produces sparse graphs. We consider the case when the Poisson process generating the weights in Equation (2) has the rate measure of a *three-parameter beta process* (3-BP) on $(0,1)$ [9, 21]:

$$\nu(dw) = \gamma \frac{\Gamma(1+\beta)}{\Gamma(1-\alpha)\Gamma(\alpha+\beta)} w^{-1-\alpha}(1-w)^{\alpha+\beta-1}\,dw, \tag{3}$$

with mass $\gamma > 0$, concentration $\beta > 0$, and discount $\alpha \in (0,1)$. In order for the 3-BP to have finite total mass $\sum_j w_j < \infty$, we require that $\beta > -\alpha$. We draw realizations of the weights from a 3-BP$(\gamma, \beta, \alpha)$ according to the stick-breaking representation given by Broderick, Jordan, and Pitman [9]. That is, the $w_i$ are the atom weights of the measure $W$ for

$$W = \sum_{i=1}^{\infty}\sum_{j=1}^{C_i} V_{i,j}^{(i)}\prod_{l=1}^{i-1}(1 - V_{i,j}^{(\ell)})\delta_{\psi_{i,j}}, \qquad\qquad C_i \overset{\text{iid}}{\sim} \text{Pois}(\gamma),$$

$$V_{i,j}^{(\ell)} \overset{\text{ind}}{\sim} \text{Beta}(1-\alpha, \beta+\ell\alpha), \qquad\qquad \psi_{i,j} \overset{\text{iid}}{\sim} B_0$$

and any continuous (i.e., non-atomic) choice of distribution $B_0$.

Since simulating an infinite number of atoms is not possible, we truncate the outer summation in $i$ to 2000 rounds, resulting in $\sum_{i=1}^{2000} C_i$ weights. The parameters of the beta process were fixed to $\gamma = 3$ and $\theta = 1$, as they do not influence the sparsity of the resulting graph frequency model, and we varied

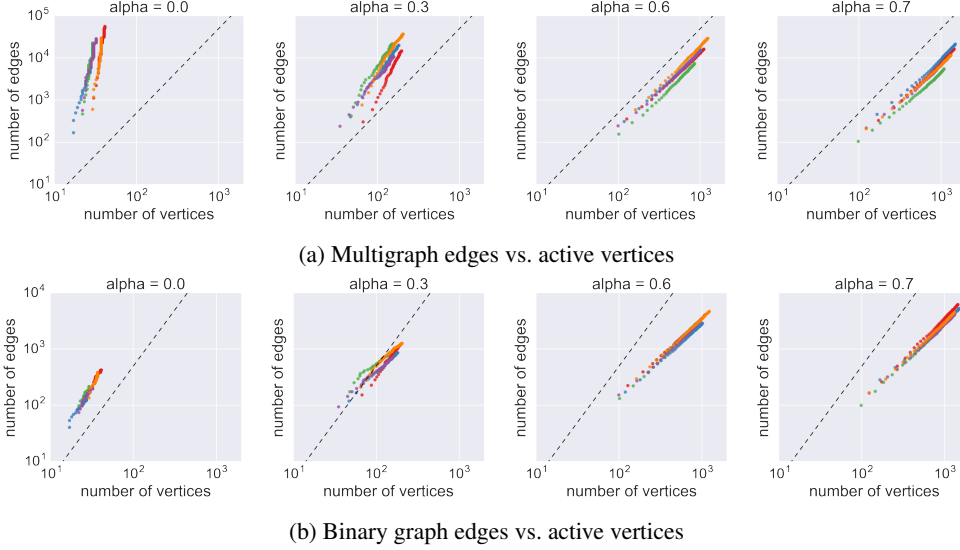

(a) Multigraph edges vs. active vertices

(b) Binary graph edges vs. active vertices

Figure 3: Data simulated from a graph frequency model with weights generated according to a 3-BP. Colors represent different random draws. The dashed line has a slope of 2.

the discount parameter $\alpha$. Given a single draw $W$ (at some specific discount $\alpha$), we then simulated the edges of the graph, where the number of Bernoulli draws $N$ varied between 50 and 2000.

Figure 3a shows how the number of edges varies versus the total number of active vertices for the multigraph, with different colors representing different random seeds. To check whether the generated graph was sparse, we determined the exponent by examining the slope of the data points (on a log-scale). In all plots, the black dashed line is a line with slope 2. In the multigraph, we found that for the discount parameter settings $\alpha = 0.6, 0.7$, the slopes were below 2; for $\alpha = 0, 0.3$, the slopes were greater than 2. This corresponds to our theoretical results; for $\alpha < 0.5$ the multigraph is dense with slope greater than 2, and for $\alpha > 0.5$ the multigraph is sparse with slope less than 2. Furthermore, the sparse graphs exhibit *power law* relationships between the number of edges and vertices, i.e., $|E_N| \overset{\text{a.s.}}{\sim} c \, |V_N|^b$, $N \to \infty$, where $b \in (1, 2)$, as suggested by the linear relationship in the plots between the quantities on a log-scale. Note that there are necessarily fewer edges in the binary graph than in the multigraph, and thus this plot implies that the binary graph frequency model can also capture sparsity. Figure 3b confirms this observation; it shows how the number of edges varies with the number of active vertices for the binary graph. In this case, across $\alpha \in (0, 1)$, we observe slopes that are less than 2. This agrees with our theory from Section 5, which states that the binary graph is sparse for any $\alpha \in (0, 1)$.

# 7 Conclusions

We have proposed an alternative form of exchangeability for random graphs, which we call *edge exchangeability*, in which the distribution of a graph sequence is invariant to the order of the edges. We have demonstrated that edge-exchangeable graph sequences, unlike traditional vertex-exchangeable sequences, can be sparse by developing a class of edge-exchangeable graph frequency models that provably exhibit sparsity. Simulations using edge frequencies drawn according to a three-parameter beta process confirm our theoretical results regarding sparsity. Our results suggest that a variety of future directions would be fruitful—including theoretically characterizing different types of power laws within graph frequency models, characterizing the use of truncation within graph frequency models as a means for approximate Bayesian inference in graphs, and understanding the full range of distributions over sparse, edge-exchangeable graph sequences.

**Acknowledgments**

We would like to thank Bailey Fosdick and Tyler McCormick for helpful conversations.

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
