[Supplementary Material]

# Edge-exchangeable graphs and sparsity: supplementary material

**Diana Cai**
Dept. of Statistics, U. Chicago
Chicago, IL 60637
dcai@uchicago.edu

**Trevor Campbell**
CSAIL, MIT
Cambridge, MA 02139
tdjc@mit.edu

**Tamara Broderick**
CSAIL, MIT
Cambridge, MA 02139
tbroderick@csail.mit.edu

## A   Overview

In Appendix B, we provide more examples of graph models that are either vertex exchangeable or Kallenberg exchangeable. In Appendix C, we establish characterizations of edge exchangeability in graphs via existing notions of exchangeability for combinatorial structures such as random partitions and feature allocations. In Appendix D, we provide full proof details for the theoretical results in the main text.

## B   More exchangeable graph models

Many popular graph models are vertex exchangeable. These models include the classic Erdős–Rényi model [11], as well as Bayesian generative models for network data, such as the stochastic block model [17], the mixed membership stochastic block model [1], the infinite relational model [18, 28], the latent space model [16], the latent feature relational model [22], the infinite latent attribute model [24], and the random function model [21]. See Orbanz and Roy [23] and Lloyd et al. [21] for more examples and discussion.

Recently, a number of extensions to the Kallenberg-exchangeable model of Caron and Fox [9], which builds on early work on bipartite graphs by Caron [8], have also been developed. These models include extensions to stochastic block models [15], mixed membership stochastic block models [27], and dynamic network models [25].

## C   Characterizations of edge-exchangeable graph sequences

We introduced edge exchangeability, a new notion of exchangeability for graphs. Just as the Aldous-Hoover theorem provides a characterization of the distribution of vertex-exchangeable graphs, it is desirable to provide a characterization of edge exchangeability in graphs. Below we show how characterization theorems that already exist for other combinatorial structures can be readily applied to provide characterizations for edge exchangeability in graphs.

We first develop mappings from edge-exchangeable graph sequences to familiar combinatorial structures—such as partitions [26], feature allocations [4], and trait allocations [6, 7]—showing that edge exchangeability in the graph corresponds to exchangeability in those structures. In this manner, we provide characterizations of the case where one edge is added to the graph per step in Appendix C.1.1, where multiple unique edges may be added per step in Appendix C.1.2, and where multiple (non)unique edges may be added in Appendix C.1.3.

A limitation of these connections is that it is not immediately clear how to recover the connectivity in the graph from the mapped combinatorial object; for instance, given a particular feature allocation, the graph to which it corresponds is not identifiable. This issue has been addressed in a purely combinatorial context via *vertex allocations* and the *graph paintbox* [7] using the general theory of

trait allocations. In Appendix C.2, we provide an alternative connection to *ordered* combinatorial structures [4, 7] under the assumption that vertex labels are provided. This assumption is often reasonable in the setting of network data where the vertices and edges are observed directly. By contrast, it is unusual to assume that labels are provided for blocks in the case of partitions, feature allocations, and trait allocations since, in these cases, the combinatorial structure is typically entirely latent in real data analysis problems. For instance, in clustering applications, finding parameters that describe each cluster is usually part of the inference problem. In the graph case, though, the use of an ordered structure identifies the particular pair of vertices corresponding to each edge in the graph, allowing recovery of the graph itself.

## C.1 The step collection sequence and connections to other forms of combinatorial exchangeability

In order to analyze edge-exchangeable graphs using the existing combinatorial machinery of random partitions, feature allocations, and trait allocations, we introduce a new combinatorial structure, the step collection sequence, which can take the form of a sequence of partitions, feature allocations, or trait allocations. As we will now see, the step collection sequence can be constructed from the step-augmented graph sequence in the following way.

Suppose we assign a unique label $\phi$ to each pair of vertices. Then if a pair of vertices is labeled $\phi$, we may imagine that any particular edge between this pair of vertices is assigned label $\phi$ when it appears. Let $\phi_j$ be the $j$th such unique edge label.

Recall that we consider a sequence of graphs defined by its step-augmented edge sequence $E_n'$. Let $S_j$ be the set of steps up to the current step $n$ in which any edge labeled $\phi_j$ was added. If $m$ edges labeled $\phi_j$ were added in a single step $s$, $s$ appears in $S_j$ with multiplicity $m$. So each element $s \in S_j$ is an element of $[n]$. Let $K_n$ be the number of unique vertex pairs seen among edges introduced up until the current step $n$. Then we may define $C_n$ to be the collection of step sets across edges that have appeared by step $n$:

$$C_n = \{S_1, \ldots, S_{K_n}\}.$$

Finally, we can define the *step collection sequence* $C = (C_1, C_2, \ldots)$ as the sequence of $C_n$ for $n = 1, 2, \ldots$.. Note that it is not clear how to recover the original edge connectivity of the graph from the step collection sequence, or whether it is possible to modify the sequence (or the labels $\phi_j$) such that it is easy to recover connectivity while maintaining the (non-trivial) connections to combinatorial exchangeability provided in Appendices C.1.1 to C.1.3 below.

**Example C.1.** Suppose we have the edge sequence

$$E_1 = \{\{2,3\}, \{3,6\}\},$$
$$E_2 = \{\{2,3\}, \{3,6\}\},$$
$$E_3 = \{\{2,3\}, \{3,6\}, \{6,6\}, \{3,6\}\},$$
$$E_4 = \{\{2,3\}, \{1,4\}, \{3,6\}, \{6,6\}, \{3,6\}\},$$

with step-augmentation

$$E_4' = \{(\{2,3\}, 1), (\{1,4\}, 4), (\{3,6\}, 1), (\{6,6\}, 3), (\{3,6\}, 3)\}$$

for $E_4$. Now we label the unique edges in $E_n'$. Using an order of appearance scheme [4] to index the labels, $E_4'$ becomes

$$\{(\phi_1, 1), (\phi_2, 1), (\phi_3, 3), (\phi_1, 3), (\phi_4, 4)\},$$

where the labels $\phi_j$ correspond to the four unique vertex pairs: $\phi_1 = \{3,6\}, \phi_2 = \{2,3\}, \phi_3 = \{6,6\}, \phi_3 = \{1,4\}$. The step collection sequence for $C_1, \ldots, C_4$ is

$$C_1 = \{\underbrace{\{1\}}_{\phi_1}\}, \quad C_2 = \{\underbrace{\{1\}}_{\phi_1}\}, \quad C_3 = \{\underbrace{\{1,3\}}_{\phi_1}, \underbrace{\{3\}}_{\phi_3}\}, \quad C_4 = \{\underbrace{\{1,3\}}_{\phi_1}, \underbrace{\{3\}}_{\phi_3}, \underbrace{\{4\}}_{\phi_4}\}.$$

Here each element of $C_n$ is a set corresponding to one of the four unique labels $\phi_j$ and contains all step indices up to step $n$ in which an edge with that label was added to the graph sequence. ∎

(a) Partition

(b) Feature allocation

(c) Trait allocation

Figure 1: Connection of edge-exchangeable graphs with partitions, feature allocations, and trait allocations. Light blocks represent 0, dark blocks either represent 1 or the specified count. In a partition, exactly one edge arrives in each step. In a feature allocation, multiple edges may arrive at each step, but at most one edge arrives between any two vertices at each step. In a trait allocation, there may be multiple edges of any type.

To see that the step collection sequence can be interpreted as a familiar combinatorial object, we recall the following definitions. A *partition* $C_n$ of $[n]$ is a set $\{S_1, \ldots, S_{K_n}\}$ whose blocks, or *clusters*, are mutually exclusive, i.e., $S_i \cap S_j = \emptyset, i \neq j$, and exhaustive, i.e., $\bigcup_j S_j = [n]$. Feature allocations relax the definition of partitions by no longer requiring the blocks to be mutually exclusive and exhaustive. A *feature allocation* $C_n$ of $[n]$ is a multiset $\{S_1, \ldots, S_{K_n}\}$ of subsets of $[n]$, such that any datapoint in $[n]$ occurs in finitely many *features* $S_j$ [4]. A *trait allocation* generalizes the feature allocation where now each $S_j$, called a *trait*, may itself be a multiset [6, 7].

We see that the step collection $C_n$ can be interpreted as follows. If a single edge is added to the graph at each round, $C_n$ is a partition of $[n]$, and the step collection sequence is a projective partition sequence. If at most one edge is added between any pair of vertices at each step, $C_n$ is a feature allocation of $[n]$, and the step collection sequence is a projective sequence of feature allocations. In the most general case, when multiple edges may be added between any pair of vertices at each step, $C_n$ is a trait allocation of $[n]$, and the step collection sequence is a projective sequence of trait allocations.

In the following examples, corresponding to Figure 1, we show different step collection sequences that correspond to a partition, a feature allocation, and a trait allocation.

**Example C.2** (Partition). Consider the step collection $C_5 = \{\{1, 3\}, \{2\}, \{4\}, \{5\}\}$. The edges form a partition of the steps. Here exactly one edge arrives in each step. ∎

**Example C.3** (Feature allocation). Consider the step collection $C_5 = \{\{1, 3\}, \{1\}, \{1, 5\}, \{3, 4\}\}$. This step collection forms a feature allocation of the steps. Thus in this case, there may be multiple *unique* edges arriving in each step. ∎

**Example C.4** (Trait allocation). In a trait allocation, there may be multiple edges (not necessarily unique) at each step. Consider the step collection $C_5 = \{\{1, 3, 3, 3\}, \{1\}, \{1, 5\}, \{3\}, \{4, 4\}\}$. This collection forms a trait allocation of the steps, where elements of $C_5$ are now multisets. ∎

In this section, we have connected certain types of edge-exchangeable graphs to partitions and feature allocations. In the next two sections, we make use of known characterizations of these combinatorial objects to characterize edge exchangeability in graphs.

### C.1.1 Partition connection

First consider the connection to partitions. In this case, suppose that each index in $[n]$ appears exactly once across all of the subsets of $C_n$. This assumption on $C_n$ is equivalent to assuming that in the original graph sequence $E_1, E_2, \ldots$, we have that $E_{n+1}$ always has exactly one more edge than $E_n$. In this case, $C_n$ is exactly a *partition* of $[n]$; that is, $C_n$ is a set of mutually exclusive and exhaustive subsets of $[n]$. If the edge sequence $(E_n)$ is random, then $(C_n)$ is random as well.

We say that a partition sequence $C_1, C_2, \ldots$, where $C_n$ is a (random) partition of $[n]$ and $C_m \subseteq C_n$ for all $m \leq n$, is infinitely exchangeable if, for all $n$, permuting the indices in $[n]$ does not change the distribution of the (random) partitions [26]. Permuting the indices $[n]$ in the partition sequence $(C_m)$

corresponds to permuting the order in which edges are added in our graph sequence $(E_m)$. As an example of a model that generates a step collection sequence corresponding to a partition sequence, consider the frequency model we introduced in Section 3 where the weights are normalized. At each step, we choose a single edge according the resulting probability distribution over pairs of vertices.

Given this connection to exchangeable partitions, the *Kingman paintbox theorem* [20] provides a characterization of edge exchangeability in graph sequences that introduce one edge per step: in particular, it guarantees that a graph sequence that adds exactly one edge per step is edge exchangeable if and only if the associated step collection sequence $(C_n)$ has a Kingman paintbox representation. An alternate characterization of edge exchangeability in graph sequences that introduce one edge per step is provided by *exchangeable partition probability functions (EPPFs)* [26]. In particular, a graph sequence that introduces one edge per step is edge-exchangeable if and only if the marginal distribution of $C_n$ (the step collection at step $n$) is given by an EPPF for all $n$.

### C.1.2 Feature allocation connection

Next we notice that it need not be the case that exactly one edge is added at each step of the graph sequence, e.g. between $E_n$ and $E_{n+1}$. If we allow multiple unique edges at any step, then the step collection $C_n$ is just a set of subsets of $[n]$, where each subset has at most one of each index in $[n]$. Suppose that any $m$ belongs to only finitely many subsets in $C_n$ for any $n$. That is, we suppose that only finitely many edges are added to the graph at any step. Then $C_n$ is an example of a *feature allocation* [4]. Again, if $(E_n)$ is random, then $(C_n)$ is random as well.

We say that a (random) feature allocation sequence $(C_m)$ is infinitely exchangeable if, for any $n$, permuting the indices of $[n]$ does not change the distribution of the (random) feature allocations [3, 4]. Permuting the indices $[n]$ in the sequence $(C_m)$ corresponds to permuting the steps when edges are added in the edge sequence $(E_m)$. Consider the following example of a graph frequency model that produces a step collection sequence corresponding to an exchangeable feature allocation. For $n = 1, 2, \ldots$, we draw whether the graph has an edge $\{i, j\}$ at time step $n$ as Bernoulli with probability $w_{\{i,j\}} = w_i w_j$. Thus, in each step, we draw at most one edge per unique vertex pair. But we may draw multiple edges in the same step.

Similarly to the partition case in Section C.1.1, we can apply known results from feature allocations to characterize edge exchangeability in graph models of this form. For instance, we know that the *feature paintbox* [4, 7] characterizes distributions over exchangeable feature allocations (and therefore the step collection sequence for graphs of this form) just as the Kingman paintbox characterizes distributions over exchangeable partitions (and therefore the step collection sequence for edge-exchangeable graphs with exactly one new edge per step).

We may also consider feature paintbox distributions with extra structure. For instance, the step collection sequence is said to have an *exchangeable feature probability function* (EFPF) [4] if the probability of each step collection $C_n$ in the sequence can be expressed as a function only of the total number of steps $n$ and the subset sizes within $C_n$ (i.e. the edge multiplicities in the graph), and is symmetric in the subset sizes. As another example, the step collection sequence is said to have a *feature frequency model* if there exists a (random) sequence of probabilities $(w_j)_{j=1}^{\infty}$ associated with edges $j = 1, 2, \ldots$ and a number $\lambda > 0$, conditioned on which the step collection sequence arises from the graph built by adding edge $j$ at each step independently[1] with probability $w_j$ for all values of $j \in \mathbb{N}$, along with an additional $\mathrm{Poiss}(\lambda)$ number of edges that never share a vertex with any other edge in the sequence. In other words, the graph is constructed with a graph frequency model as in the main text of the present work (modulo the aforementioned additional Poisson number of edges). Theorem 17 ("Equivalence of EFPFs and feature frequency models") from [4] shows that these two examples are actually equivalent: if the step collection sequence has an EFPF, it has a feature frequency model, and vice versa.

### C.1.3 Further extensions

Finally, we may consider the case where at every step, any non-negative (finite) number of edges may be added *and* those edges may have non-trivial (finite) multiplicity; that is, the multiplicity of any edge at any step can be any non-negative integer. By contrast, in Section C.1.2, each unique edge occurred at most once at each step. In this case, the step collection $C_n$ is a set of subsets of $[n]$. The

subsets need not be unique or exclusive since we assume any number of edges may be added at any step. And the subsets themselves are multisets since an edge may be added with some multiplicity at step $n$. We say that $C_n$ is a *trait allocation*, which we define as a generalization of a feature allocation where the subsets of $C_n$ are multisets. As above, if $(E_n)$ is random, $(C_n)$ is as well.

We say that a (random) trait allocation sequence $(C_m)$ is infinitely exchangeable if, for any $n$, permuting the indices of $[n]$ does not change the distribution of the (random) trait allocation. Here, permuting the indices of $[n]$ corresponds to permuting the steps when edges are added in the edge sequence $(E_m)$. A graph frequency model that generates a step collection sequence as a trait allocation sequence is the multiple-edge-per-step frequency model sampling procedure described in Section 3. Here, at each step, multiple edges can appear each with multiplicity potentially greater than 1, requiring the full generality of a trait allocation sequence.

Campbell et al. [7] characterize exchangeable trait allocations via, e.g., probability functions and paintboxes and thereby provide a characterization over the corresponding step collection sequences of such edge-exchangeable graphs.

## C.2 Connections to exchangeability in ordered combinatorial structures

As noted earlier, it is not immediately clear how to recover the connectivity in an edge-exchangeable graph from the step collection sequence, nor how to do so in a way that preserves non-trivial connections to other exchangeable combinatorial structures. Campbell et al. [7] considers an alternative to the step collection sequence in which the (multi)subsets in the combinatorial structure correspond to *vertices* rather than edges, known as a *vertex allocation*. This allows for the characterization of edge-exchangeable graphs via the *graph paintbox* using the general theory of trait allocations, while maintaining an explicit representation of the structure of the graph, i.e., the connection between edges that share a vertex.

If we are willing to eschew the unordered nature of the step collection sequence, and assume that we have an a priori labeling on the vertices, there is yet another alternative using the *ordered step collection sequence*. The availability of labeled vertices is often a reasonable assumption in the setting of network data, where the vertices and edges are typically observed directly. Suppose the vertices are labeled using the natural numbers $1, 2, \ldots$. Then we can use the ordering of the vertex labels to order the vertex pairs in a diagonal manner, i.e. $\{1, 1\}, \{1, 2\}, \{2, 2\}, \{1, 3\}, \{2, 3\}, \ldots$. Note that, for the purpose of building this diagonal ordering, we consider the lowest-valued index in each vertex pair first. We build the step collection sequence $(C_n)$ in the same manner as before, except that each step collection $C_n$ is no longer an unordered collection of subsets; the subsets derive their order from the vertex pairs they represent. For example, if we observe edges at vertex pairs $\{1, 1\}$ and $\{1, 2\}$ at step 1, and edges at vertex pairs $\{1, 1\}$ and $\{2, 3\}$ at step 2, then

$$C_1 = (\{1\}, \{1\}, \emptyset, \emptyset, \dots)$$

and

$$C_2 = (\{1, 2\}, \{1\}, \emptyset, \emptyset, \{2\}, \emptyset, \dots).$$

Since we know the order of the subsets in each $C_n$ as they relate to the vertex pairs in the graph and their connectivity, we can recover the graph sequence from the ordered step collection sequence $(C_n)$. Exchangeability in an ordered step collection sequence means that the distribution is invariant to permutations of the indices within the subsets (although the ordering of the subsets themselves cannot be changed). Given this notion of exchangeability, the earlier connections to exchangeable partitions, feature allocations, and trait allocations remain true, modulo the fact that they must themselves be ordered. Broderick et al. [4] provides a paintbox characterization of ordered exchangeable feature allocations, thereby providing characterizations (via the earlier connections to partitions and feature allocations) of edge-exchangeable graphs that add either one or multiple unique edges per step. Note that, in these cases, this is a full characterization of edge-exchangeable graphs, by contrast to Appendix C.1, where we provided a characterization only of edge exchangeability in graphs. We suspect that a similar characterization of edge-exchangeable graphs with multiple (non)unique edges per step is available by examining characterizations of exchangeable ordered trait allocations.

# D  Proofs

The proof of the main theorem in the paper (Theorem 5.3) follows from a collection of lemmas below. Lemma 5.2 characterizes the expected number of vertices and edges; Lemma D.3 establishes a useful transformation of those expectations; and Lemma D.4 shows that the two sets of expectations are asymptotically equivalent, so it is enough to consider the transformed expectation. Lemma D.6 provides the asymptotics of the transformed expectations. Finally, Lemma 5.1 shows that the random sequences converge almost surely to their expectations, yielding the final result.

## D.1  Preliminaries

**Notation.**  We first define the asymptotic notation used in the main paper and appendix. We use the notation "a.s." to mean almost surely, or with probability 1. Let $(X_n)_{n \in \mathbb{N}}, (Y_n)_{n \in \mathbb{N}}$ be two random sequences. We say that $X_n \overset{\text{a.s.}}{=} O(Y_n)$ if $\limsup_{n \to \infty} \frac{X_n}{Y_n} < \infty$ a.s., and that $X_n \overset{\text{a.s.}}{=} \Omega(Y_n)$ if $Y_n \overset{\text{a.s.}}{=} O(X_n)$ a.s. We say that $X_n \overset{\text{a.s.}}{=} o(Y_n)$ if $\lim_{n \to \infty} \frac{X_n}{Y_n} = 0$ a.s. Lastly, we say that $X_n \overset{\text{a.s.}}{=} \Theta(Y_n)$ if $X_n \overset{\text{a.s.}}{=} O(Y_n)$ and $Y_n \overset{\text{a.s.}}{=} O(X_n)$.

Let $V_n, E_n$ be the respective sets of active vertices and edges at step $n$ in the multigraph, and $|V_n|, |E_n|$ be their respective cardinalities, as defined in the main text. We use the notation $\bar{V}_n$ and $\bar{E}_n$ to represent these analogous vertex and edge sets for the binary graph. Note that $\bar{V}_n$ is the same as $V_n$.

**Useful results.**  We present two useful theorems for analyzing expectations involving random sums of functions of points from Poisson point processes. Below, we will apply these theorems repeatedly to get expectations of graph quantities. The first theorem is Campbell's theorem, which is used to compute the moments of functionals of a Poisson process. We state it below for completeness, and refer to Kingman [19, Sec. 3.2] for details.

**Theorem D.1** (Campbell's theorem). *Let $\Pi$ be a Poisson point process on $S$ with rate measure $\nu$, and let $f : S \to \mathbb{R}$ be measurable. If $\int_S \min(|f(x)|, 1)\, \nu(dx) < \infty$, then*

$$\mathbb{E}\left(\exp\left(c \sum_{x \in \Pi} f(x)\right)\right) = \exp\left(\int_S (\exp(cf(x)) - 1)\, \nu(dx)\right)$$

*for any $c \in \mathbb{C}$, and furthermore,*

$$\mathbb{E}\left(\sum_{x \in \Pi} f(x)\right) = \int_S f(x)\, \nu(dx).$$

The second theorem is a specific form of the Slivnyak-Mecke theorem, which is useful for computing the expected sum of a function of each point $x \in \Pi$ and $\Pi \setminus \{x\}$ over all points in a Poisson point process $\Pi$. If each point in $\Pi$ is thought of as relating to a particular vertex in a graph, the Slivnyak-Mecke theorem allows us to take expectations of the sum (over all possible vertices in the graph) of a function of each vertex and all its possible edges. For example, it is used below to compute the expected number of active vertices by taking the expected sum of vertices that have nonzero degree. We state it below for completeness, and refer to Daley and Vere-Jones [10, Prop. 13.1.VII] and Baddeley et al. [2, Thm. 3.1,Thm. 3.2] for details.

**Theorem D.2** (Slivnyak-Mecke theorem). *Let $\Pi$ be a Poisson point process on $S$ with rate measure $\nu$, and let $f : S \times \Omega \to \mathbb{R}_+$ be measurable. Then*

$$\mathbb{E}\left(\sum_{x \in \Pi} f(x, \Pi \setminus \{x\})\right) = \int_S \mathbb{E}\left(f(x, \Pi)\right) \nu(dx).$$

## D.2  Graph moments

In this section, we give the expected number of vertices and expected number of edges for the multi- and binary graph cases. We begin by defining the degree $D_i$ of vertex $i$ in the multigraph and the

degree $\bar{D}_i$ of vertex $i$ in the binary graph, respectively, as

$$D_i = \sum_j M_{\{i,j\}} \qquad\qquad \bar{D}_i = \sum_j \mathbb{1}\left(M_{\{i,j\}} > 0\right). \qquad (\text{D.1})$$

Now we present the expected number of edges and vertices. We note that both the multi- and binary graphs have the same number of (active) vertices, and so their expectations are the same.

**Lemma** (5.2, main text). *The expected number of vertices and edges for the multi- and binary graphs are*

$$\mathbb{E}\left(|\bar{V}_n|\right) = \mathbb{E}\left(|V_n|\right) = \int \left[1 - \exp\left(-\int 1 - (1 - wv)^n \, \nu(\mathrm{d}v)\right)\right] \nu(\mathrm{d}w),$$

$$\mathbb{E}\left(|E_n|\right) = \frac{n}{2} \iint wv \, \nu(\mathrm{d}w)\,\nu(\mathrm{d}v),$$

$$\mathbb{E}\left(|\bar{E}_n|\right) = \frac{1}{2} \iint (1 - (1 - wv)^n)\, \nu(\mathrm{d}w)\,\nu(\mathrm{d}v).$$

*Proof.* Using the tower property of conditional expectation and Fubini's theorem, we have that the expected number of vertices is

$$\mathbb{E}\left(|V_n|\right) = \mathbb{E}\left(\mathbb{E}\left(\sum_i \mathbb{1}(D_i > 0)\,\middle|\,\mathcal{W}\right)\right) = \mathbb{E}\left(\sum_i \mathbb{P}\left(D_i > 0\,\middle|\,\mathcal{W}\right)\right),$$

followed by the definition of degree in Equation (D.1) and the binomial density,

$$\mathbb{E}\left(|V_n|\right) = \mathbb{E}\left(\sum_i \left[1 - \prod_j \mathbb{P}\left(M_{\{i,j\}} = 0\,|\,\mathcal{W}\right)\right]\right) = \mathbb{E}\left(\sum_{w \in \mathcal{W}} \left[1 - \prod_{v \in \mathcal{W}\setminus\{w\}} (1 - wv)^n\right]\right).$$

Using the Slivnyak-Mecke theorem (Theorem D.2),

$$\mathbb{E}\left(|V_n|\right) = \int \mathbb{E}\left(1 - \prod_{v \in \mathcal{W}} (1 - wv)^n\right) \nu(\mathrm{d}w)$$

$$= \int \left[1 - \mathbb{E}\left(\exp\left(n \sum_{v \in \mathcal{W}} \log(1 - wv)\right)\right)\right] \nu(\mathrm{d}w),$$

and finally by Campbell's theorem (Theorem D.1) on the inner expectation,

$$\mathbb{E}\left(|V_n|\right) = \int \left[1 - \exp\left(-\int (1 - (1 - wv)^n)\,\nu(\mathrm{d}v)\right)\right] \nu(\mathrm{d}w).$$

For the expected number of edges, we can again apply the tower property and Fubini's theorem followed by repeated applications of Slivnyak-Mecke to the expectations to get:

$$\mathbb{E}(|E_n|) = \mathbb{E}\left(\mathbb{E}\left(\frac{1}{2}\sum_{i \neq j} M_{\{i,j\}}\,\middle|\,\mathcal{W}\right)\right) = \frac{1}{2}\int \mathbb{E}\left(\sum_{v \in \mathcal{W}} nwv\right)\nu(\mathrm{d}w) = \frac{n}{2}\int wv\nu(\mathrm{d}w)\nu(\mathrm{d}v).$$

The expected number of edges for the binary case is obtained similarly via Fubini and Slivnyak-Mecke:

$$\mathbb{E}(|\bar{E}_n|) = \mathbb{E}\left(\frac{1}{2}\sum_{i \neq j} P(M_{\{i,j\}} > 0|\mathcal{W})\right) = \frac{1}{2}\mathbb{E}\left(\sum_{w \in \mathcal{W}, v \in \mathcal{W}\setminus\{w\}} (1 - (1 - wv)^n)\right)$$

$$= \frac{1}{2}\int\int (1 - (1 - wv)^n)\,\nu(\mathrm{d}w)\,\nu(\mathrm{d}v).$$

$\square$

The asymptotic behavior of these quantities is difficult to derive directly due to the discreteness of the indices $n$. Therefore, we rely on a technique called *Poissonization*, which allows us to bypass this difficulty by instead considering a continuous analog of the quantities in order to get asymptotic behaviors. Below, we introduce primed notation $V_t', E_t', \bar{E}_t', D_{t,i}'$ to represent the Poissonized quantities for the vertices, multigraph edges, binary edges, and the degree of a vertex, where the index $t$ now represents a continuous quantity. These will be defined such that $V_N'$ has the same asymptotic behavior as $V_N$, $E_N'$ has the same asymptotic behavior as $E_N$, and so on.

Given $\mathcal{W}$, let $\Pi_{ij}$ be the Poisson process generated with rate $w_i w_j$ if $i < j$ and rate 0 if $i = j$, and let $\Pi_{ji} = \Pi_{ij}$. Let $\Pi_i := \bigcup_{j=1}^{\infty} \Pi_{ij}$, which is a Poisson process with rate $u_i := \sum_{j:j\neq i} w_i w_j$ via Poisson process superposition [19, Sec. 2.2]. If we think of $t$ as continuous time passing, the process $\Pi_{ij}$ represents the times at which new edges are added between vertices $i$ and $j$, and $\Pi_i$ represents the times at which any new edges involving vertex $i$ are added.

Thus, we define the Poissonized degree of vertex $i$ in the multi- and binary graph cases, respectively, to be a function of the continuous parameter $t > 0$,

$$D_{t,i}' = |\Pi_i \cap [0,t]|, \qquad \bar{D}_{t,i}' = \sum_j \mathbb{1}\left(|\Pi_{ij} \cap [0,t]| > 0\right).$$

We can define the Poissonized graph quantities of interest using these two quantities:

$$|\bar{V}_t'| = |V_t'| = \sum_i \mathbb{1}(D_{t,i}' > 0), \qquad |E_t'| = \frac{1}{2}\sum_{i=1}^{\infty} D_{t,i}', \qquad |\bar{E}_t'| = \frac{1}{2}\sum_i \bar{D}_{t,i}'.$$

**Lemma D.3.** *The expected number of Poissonized vertices and edges for the multi- and binary graphs is*

$$\mathbb{E}\left(|V_t'|\right) = \int \left[1 - \exp\left(-\int (1 - e^{-twv})\,\nu(dv)\right)\right]\nu(dw)$$

$$\mathbb{E}\left(|E_t'|\right) = \frac{t}{2}\iint wv\,\nu(dw)\,\nu(dv)$$

$$\mathbb{E}\left(|\bar{E}_t'|\right) = \frac{1}{2}\iint (1 - \exp(-twv))\,\nu(dw)\,\nu(dv).$$

*Proof.* For the expected number of Poissonized vertices, we apply the tower property and Fubini's theorem to get

$$\mathbb{E}\left(|V_t'|\right) = \mathbb{E}\left(\mathbb{E}\left(\sum_i \mathbb{1}(D_{t,i}' > 0)\,\middle|\,\mathcal{W}\right)\right) = \mathbb{E}\left(\sum_i 1 - \mathbb{P}\left(D_{t,i} = 0\,|\,\mathcal{W}\right)\right).$$

Using the fact that $D_{t,i}'|\mathcal{W}$ is Poisson-distributed,

$$\mathbb{E}\left(|V_t'|\right) = \mathbb{E}\left(\sum_i 1 - \exp\left(-tu_i\right)\right) = \mathbb{E}\left(\sum_{w\in\mathcal{W}} 1 - \exp\left(-tw\sum_{v\in\mathcal{W}\setminus\{w\}} v\right)\right).$$

Finally, by the Slivnyak-Mecke theorem and Campbell's theorem,

$$\mathbb{E}\left(|V_t'|\right) = \int \mathbb{E}\left(1 - \exp\left(-tw\sum_{v\in\mathcal{W}} v\right)\right)\nu(dw)$$

$$= \int \left[1 - \exp\left(\int (e^{-twv} - 1)\,\nu(dv)\right)\right]\nu(dw).$$

For the expected number of Poissonized edges, after applying Fubini's theorem and Slivnyak-Mecke we have

$$\mathbb{E}\left(|E_t'|\right) = \mathbb{E}\left(\frac{1}{2}\sum_i D_{t,i}'\right) = \mathbb{E}\left(\frac{1}{2}\sum_i \mathbb{E}\left(D_{t,i}'|\mathcal{W}\right)\right)$$

$$= \mathbb{E}\left(\frac{1}{2}\sum_i u_i\right) = \mathbb{E}\left(\frac{1}{2}\sum_{w\in\mathcal{W}, v\in\mathcal{W}\setminus\{w\}} wv\right)$$

$$= \frac{1}{2}\int\int wv\,\nu(dw)\,\nu(dv).$$

For the expected number of Poissonized edges in the binary case, noting that $|\Pi_{ij}\cap[0,t]|$ is Poisson-distributed with rate $tw_iw_j$, and applying Fubini's theorem and Slivnyak-Mecke, we have:

$$\mathbb{E}(|\bar{E}_t'|) = \mathbb{E}\left(\mathbb{E}\left(\sum_i \bar{D}_{t,i}'|\mathcal{W}\right)\right) = \mathbb{E}\left(\sum_{w\in\mathcal{W}, v\in\mathcal{W}\setminus\{w\}}(1-\exp(-twv))\right)$$

$$= \int\int(1-\exp(-twv))\,\nu(dw)\,\nu(dv).$$

$\square$

### D.3 Asymptotics

We have defined the expected number of vertices and edges for the multigraph and binary graph cases (Lemma 5.2) and presented the Poissonized version of these expectations (Lemma D.3). We now show in Lemma D.4 that the expected graph quantities and their Poissonized expectations are asymptotically equivalent.

**Lemma D.4.** *The Poissonized expectations for the number of vertices and the number of edges in the multi- and binary graphs are asymptotically equivalent to the original expectations; i.e., as $n\to\infty$,*

$$\mathbb{E}\left(|V_n'|\right) \sim \mathbb{E}\left(|V_n|\right),$$
$$\mathbb{E}\left(|E_n'|\right) \sim \mathbb{E}\left(|E_n|\right),$$
$$\mathbb{E}\left(|\bar{E}_n'|\right) \sim \mathbb{E}\left(|\bar{E}_n|\right).$$

*Proof.* For the vertices, we have

$$\mathbb{E}\left(|V_n|-|V_n'|\right) = \int\left[\exp\left(-\int(1-e^{-nwv})\,\nu(dv)\right) - \exp\left(-\int(1-(1-wv)^n)\,\nu(dv)\right)\right]\nu(dw).$$

Using the elementary inequalities

$$0\le e^{-nx}-(1-x)^n \le nx^2e^{-nx}, \qquad x\in[0,1],\ n>0$$
$$0\le e^{-a}-e^{-b}\le b-a, \qquad 0\le a\le b,$$

we have

$$0\le \mathbb{E}\left(|V_n|-|V_n'|\right) \le \int\int n(wv)^2e^{-nwv}\,\nu(dv)\,\nu(dw). \tag{D.2}$$

Finally, note that

$$\forall n>0, \forall w,v\in[0,1], \quad nwve^{-nwv}\le e^{-1}$$

and

$$\int\int e^{-1}wv\,\nu(dw)\,\nu(dv) = e^{-1}\left(\int w\,\nu(dw)\right)^2 < \infty.$$

Therefore by Lebesgue dominated convergence,

$$0\le \lim_{n\to\infty}\mathbb{E}\left(|V_n|-|V_n'|\right) \le \int\int \lim_{n\to\infty} n(wv)^2e^{-nwv}\,\nu(dv)\,\nu(dw) = 0,$$

so we have that $\lim_{n\to\infty} \mathbb{E}\left(|V_n| - |V_n'|\right) = 0$. Since $\mathbb{E}(|V_n|), \mathbb{E}(|V_n'|)$ are monotonically increasing by inspection, $\mathbb{E}(|V_n|) \sim \mathbb{E}(|V_n'|)$, $n \to \infty$, as required.

For the binary graph edges,

$$\mathbb{E}\left(|\bar{E}_n| - |\bar{E}_n'|\right) = \frac{1}{2} \iint (\exp(-nwv) - (1-wv)^n)\, \nu(dv)\, \nu(dw).$$

Using the earlier elementary inequalities,

$$0 \le \mathbb{E}\left(|\bar{E}_n| - |\bar{E}_n'|\right) = \frac{1}{2} \iint n(wv)^2 e^{-nwv}\, \nu(dv)\, \nu(dw).$$

This is (modulo the constant factor of $1/2$) the exact expression in Equation (D.2). Therefore, the same analysis can be performed, and the result holds.

For multigraph edges,

$$\mathbb{E}\left(|E_n| - |E_n'|\right) = \frac{n}{2} \iint (wv - wv)\, \nu(dv)\, \nu(dw) = 0,$$

so $\mathbb{E}\left(|E_n|\right) \sim \mathbb{E}\left(|E_n'|\right)$, $n \to \infty$. $\qquad\square$

Lemma D.4 allows us to analyze the asymptotics of the Poissonized expectations and apply the result directly to the asymptotics of the original graph quantities. To achieve the desired asymptotics for the Poissonized expectations, we will make a further assumption on the rate measure $\nu$ generating the vertex weights in Equation (2). Namely, we assume that the tails of $\nu$ decay at a rate that will yield the appropriate weight decay in the weights $(w_j)$—and thereby the appropriate decay in vertex creation to finally yield sparsity in the graph itself. In particular, the tail of a measure $\nu$ is said to be *regularly varying* if there exists a function $\ell : \mathbb{R}_+ \to \mathbb{R}_+$ and $\alpha \in (0,1)$ such that

$$\int_x^1 \nu(dw) \sim x^{-\alpha}\ell(x^{-1}), \quad x \to 0, \qquad\qquad \forall c > 0, \lim_{x\to\infty} \frac{\ell(cx)}{\ell(x)} = 1. \qquad (D.3)$$

The condition on the function $\ell$ is equivalent to saying that $\ell$ is *slowly varying*. For additional details on regular and slow variation, see Feller [12, VIII.8]. An important equivalent formulation of Equation (D.3) that we will use in our following proof of the asymptotics of Poissonized expectations is provided by Lemma D.5 (see Gnedin et al. [14, Prop. 13] and Broderick et al. [5, Prop. 6.1] for the proof).

**Lemma D.5** (Broderick et al. [5], Gnedin et al. [14])**.** *The tail of $\nu$ is regularly varying iff there exists a function $\ell : \mathbb{R}_+ \to \mathbb{R}_+$ and $\alpha \in (0,1)$ such that*

$$\int_0^x u\nu(du) \sim x^{1-\alpha}\ell(x^{-1}), \quad x \to 0, \qquad\qquad \forall c > 0, \lim_{x\to\infty} \frac{\ell(cx)}{\ell(x)} = 1. \qquad (D.4)$$

Lemma D.5 is often easier to use than Equation (D.3) when checking whether a particular measure $\nu$ has a regularly varying tail. For example, for the three-parameter beta process, we have

$$\int_0^x u\nu(du) = \gamma \frac{\Gamma(1+\beta)}{\Gamma(1-\alpha)\Gamma(\beta+\alpha)} \int_0^x u^{-\alpha}(1-u)^{\beta+\alpha-1} du$$

$$\sim \gamma \frac{\Gamma(1+\beta)}{\Gamma(1-\alpha)\Gamma(\beta+\alpha)} \int_0^x u^{-\alpha} du, \quad x \downarrow 0$$

$$= \gamma \frac{\Gamma(1+\beta)}{\Gamma(1-\alpha)\Gamma(\beta+\alpha)} \frac{1}{1-\alpha} x^{1-\alpha},$$

so the tail of $\nu$ is regularly varying when the discount parameter $\alpha$ satisfies $\alpha \in (0,1)$ with $\ell(x^{-1})$ equal to the constant function

$$\ell(x^{-1}) = \frac{\gamma}{1-\alpha} \frac{\Gamma(1+\beta)}{\Gamma(1-\alpha)\Gamma(\beta+\alpha)}. \qquad (D.5)$$

Note that the two-parameter beta process does not exhibit this behavior (since in this case, $\alpha = 0$).

Given the two formulations of a measure $\nu$ with a regularly varying tail above, we are ready to characterize the asymptotics of the earlier Poissonized expectations.

**Lemma D.6.** *If the tail of $\nu$ is regularly varying as per Equation* (D.3)*, then as $n \to \infty$,*

$$\mathbb{E}\left(|V_n'|\right) = \Theta(n^\alpha \ell(n)), \quad \mathbb{E}\left(|E_n'|\right) = \Theta(n), \quad \mathbb{E}\left(|\bar{E}_n'|\right) = O\left(\ell(\sqrt{n}) \min\left(n^{\frac{1+\alpha}{2}}, \ell(n)n^{\frac{3\alpha}{2}}\right)\right).$$

*Proof.* Throughout this proof we use $c$ to denote a constant; the precise value of $c$ changes but is immaterial. We also define the tail of $\nu$ as $\bar{\nu}(x) := \int_x^1 \nu(dw)$, for notational brevity. Furthermore, recall that we assume the rate measure $\nu$ satisfies $\int w\nu(dw) < \infty$.

We first examine the expected number of Poissonized vertices,

$$\mathbb{E}\left(|V_n'|\right) = \int \left[1 - \exp\left(-\int(1 - e^{-nwv})\nu(dv)\right)\right] \nu(dw),$$

by splitting the integral into two parts. First, by the assumption that the tail of $\nu$ is regularly varying,

$$\int_{n^{-1}}^1 \left[1 - \exp\left(-\int(1 - e^{-nwv})\nu(dv)\right)\right] \nu(dw) \le \int_{n^{-1}}^1 \nu(dw) \sim cn^\alpha \ell(n). \qquad \text{(D.6)}$$

Next, we upper bound the integral term

$$\int_0^{n^{-1}} \left[1 - \exp\left(-\int(1 - e^{-nwv})\nu(dv)\right)\right] \nu(dw) \le \int_0^{n^{-1}} \int(1 - e^{-nwv})\nu(dv)\nu(dw)$$

$$\le \int_0^{n^{-1}} \int nwv\nu(dv)\nu(dw)$$

$$\le \left(\int v\nu(dv)\right) n \int_0^{n^{-1}} w\nu(dw)$$

$$\sim cn^\alpha \ell(n), \qquad \text{(D.7)}$$

where the asymptotic behavior in the last line follows from Lemma D.5. Thus, combining the upper bounds on Equation (D.6) and Equation (D.7) gives the bound for the entire integral: $\mathbb{E}\left(|V_n'|\right) = O(n^\alpha \ell(n))$.

Now we bound the integral below:

$$\int_{n^{-1}}^1 \left[1 - \exp\left(-\int(1 - e^{-nwv})\nu(dv)\right)\right] \nu(dw)$$

$$\ge \int_{n^{-1}}^1 \left[1 - \exp\left(-\int(1 - e^{-v})\nu(dv)\right)\right] \nu(dw)$$

$$= \left(\int_{n^{-1}}^1 \nu(dw)\right) \left(1 - \exp\left(-\int(1 - e^{-v})\nu(dv)\right)\right)$$

$$\sim cn^\alpha \ell(n),$$

where the last line follows from the assumption that the tail of $\nu$ is regularly varying. The second piece of the integral on $[0, n^{-1}]$ is bounded below by 0, and in combination, we have that $n^\alpha \ell(n) = O\left(\mathbb{E}\left(|V_n'|\right)\right)$. Now combining this with the previous upper bound result, we have $\mathbb{E}\left(|V_n'|\right) = \Theta(n^\alpha \ell(n))$.

The expected number of Poissonized multigraph edges satisfies $\mathbb{E}\left(E_n'\right) = \Theta(n)$, since

$$\mathbb{E}(|E_n'|) = \frac{n}{2} \iint wv\nu(dw)\nu(dv) = \frac{n}{2} \int w\nu(dw) \int v\nu(dv) = \frac{c^2}{2}n.$$

For the Poissonized binary graph edges, we split the integral into two pieces. We first upper bound the integral on the interval $[0, n^{-1/2}]$ and apply Theorem D.5 to get the following asymptotic behavior:

$$\frac{1}{2} \int_0^{n^{-1/2}} \int(1 - \exp(-nwv))\,\nu(dw)\,\nu(dv) \le \frac{1}{2} \int_0^{n^{-1/2}} \int nwv\,\nu(dw)\,\nu(dv)$$

$$= \frac{n}{2} \left(\int w\nu(dw)\right) \int_0^{n^{-1/2}} v\nu(dv)$$

$$\sim cn(n^{-1/2})^{1-\alpha}\ell(n^{1/2})$$

$$= cn^{\frac{1+\alpha}{2}} \ell(n^{1/2}).$$

We then bound the second portion on the interval $[n^{-1/2}, 1]$ by linearizing at $v = n^{-1/2}$. Using integration by parts and an Abelian theorem [12, Sec. XIII.5, Thm. 4] for the Laplace transform, for some constants $b, d > 0$, we have

$$\frac{1}{2} \int_{n^{-1/2}}^{1} \int (1 - \exp(-nwv)) \, \nu(\mathrm{d}w) \, \nu(\mathrm{d}v)$$

$$\leq \frac{1}{2} \int_{n^{-1/2}}^{1} \int \left( 1 - \exp\left(-n^{1/2}w\right) + nw \exp\left(-n^{1/2}w\right)(v - n^{-1/2}) \right) \nu(\mathrm{d}w) \, \nu(\mathrm{d}v)$$

$$= \frac{1}{2} \left( \int_{n^{-1/2}}^{1} \nu(\mathrm{d}v) \right) \int n^{1/2} \exp(-n^{1/2}w) \, \bar{\nu}(w) \, \mathrm{d}w$$

$$\quad + \frac{1}{2} \int_{n^{-1/2}}^{1} (nv - n^{1/2}) \, \nu(\mathrm{d}v) \int w \exp(-n^{1/2}w) \, \nu(\mathrm{d}w)$$

$$\sim bn^{\alpha} \ell^2(n^{1/2}) + \frac{1}{2} \int_{0}^{1} v \, \nu(\mathrm{d}v) \, n^{1/2} \int n^{1/2} \left( \exp(-n^{1/2}w) - n^{1/2}w \exp\left(-n^{1/2}w\right) \right) \bar{\nu}(w) \, \mathrm{d}w$$

$$\leq bn^{\alpha} \ell^2(n^{1/2}) + \frac{1}{2} \int_{0}^{1} v \, \nu(\mathrm{d}v) \, n^{1/2} \int n^{1/2} \exp(-n^{1/2}w) \, \bar{\nu}(w) \, \mathrm{d}w$$

$$\sim bn^{\alpha} \ell^2(n^{1/2}) + dn^{1/2}n^{\alpha/2}\ell(n^{1/2})$$

$$= O(n^{\frac{1+\alpha}{2}} \ell(n^{1/2})).$$

Therefore we have that $\mathbb{E}\left(|\bar{E}'_n|\right) = O(n^{\frac{1+\alpha}{2}} \ell(n^{1/2}))$.

To get the other bound, we split the integral into three pieces. First,

$$\frac{1}{2} \int_{0}^{n^{-1}} \int (1 - \exp(-nwv)) \, \nu(\mathrm{d}w) \, \nu(\mathrm{d}v)$$

$$\leq \frac{1}{2} \int_{0}^{n^{-1}} \int nwv \, \nu(\mathrm{d}w) \, \nu(\mathrm{d}v)$$

$$= \frac{n}{2} \left( \int w \, \nu(\mathrm{d}w) \right) \int_{0}^{n^{-1}} v \, \nu(\mathrm{d}v)$$

$$\sim cn(n^{-1})^{1-\alpha}\ell(n) = cn^{\alpha}\ell(n).$$

Next, integration by parts yields

$$\frac{1}{2} \int_{n^{-1/2}}^{1} \int (1 - \exp(-nwv)) \, \nu(\mathrm{d}w) \, \nu(\mathrm{d}v)$$

$$\leq \frac{1}{2} \int_{n^{-1/2}}^{1} \int (1 - \exp(-nw)) \, \nu(\mathrm{d}w) \, \nu(\mathrm{d}v)$$

$$= \frac{1}{2} \left( \int_{n^{-1/2}}^{1} \nu(\mathrm{d}v) \right) \int n \exp(-nw) \, \bar{\nu}(w) \, \mathrm{d}w$$

$$\sim c \left( n^{-1/2} \right)^{-\alpha} \ell(n^{1/2}) n^{\alpha} \ell(n)$$

$$= cn^{\frac{3\alpha}{2}} \ell(n)\ell(n^{1/2}).$$

Finally, integration by parts yields the final upper bound

$$\frac{1}{2}\int_{n^{-1}}^{n^{-1/2}}\int (1-\exp(-nwv))\,\nu(\mathrm{d}w)\,\nu(\mathrm{d}v)$$

$$\leq \frac{1}{2}\int_{n^{-1}}^{n^{-1/2}}\int (1-\exp(-n^{1/2}w))\,\nu(\mathrm{d}w)\,\nu(\mathrm{d}v)$$

$$= \frac{1}{2}\left(\int_{n^{-1}}^{n^{-1/2}}\nu(\mathrm{d}v)\right)\int n^{1/2}\exp\left(-n^{1/2}w\right)\bar{\nu}(w)\,\mathrm{d}w$$

$$\sim \left(c_1 n^{\alpha}\ell(n) - c_2 n^{\frac{\alpha}{2}}\ell(n^{1/2})\right)\left(c_3 n^{\alpha/2}\ell(n^{1/2})\right)$$

$$\sim c n^{\frac{3\alpha}{2}}\ell(n)\ell(n^{1/2}).$$

Therefore $\mathbb{E}\left(|\bar{E}_n'|\right) = O(\ell(n)\ell(n^{1/2})\,n^{\frac{3\alpha}{2}})$. $\qquad\square$

Finally, we show that $|E_n|$, $|\bar{E}_n|$, and $|V_n|$ are asymptotically equivalent to their expectations almost surely; thus, the asymptotic results for the expectation sequences applies to the random sequences.

**Lemma** (5.1, main text). *The number of edges and vertices for both the multi- and binary graphs satisfy*

$$|E_n| \overset{a.s.}{\sim} \mathbb{E}\left(|E_n|\right), \qquad |\bar{E}_n| \overset{a.s.}{\sim} \mathbb{E}\left(|\bar{E}_n|\right) \qquad |\bar{V}_n| = |V_n| \overset{a.s.}{\sim} \mathbb{E}\left(|V_n|\right), \qquad n \to \infty.$$

*Proof.* We use $X_n$ to refer to $|E_n|$, $|\bar{E}_n|$, or $|V_n|$, since the proof technique is the same for all. Since we need to show $X_n/\mathbb{E}\left(X_n\right) \overset{a.s.}{\to} 1$, by the Borel-Cantelli lemma it is sufficient to show that for any $\epsilon > 0$,

$$\sum_n P(|X_n - \mathbb{E}\left(X_n\right)| > \epsilon\mathbb{E}\left(X_n\right)) < \infty.$$

By the union bound, and the fact that $X_n$ can be expressed as a countable sum of indicators combined with the note after Theorem 4 in Freedman [13],

$$P(|X_n - \mathbb{E}\left(X_n\right)| > \epsilon\mathbb{E}\left(X_n\right))$$
$$\leq P(X_n > (1+\epsilon)\mathbb{E}\left(X_n\right)) + P(X_n < (1-\epsilon)\mathbb{E}\left(X_n\right))$$
$$\leq 2\exp\left(-\frac{\epsilon^2\mathbb{E}\left(X_n\right)}{2}\right).$$

Since $\mathbb{E}(X_n) \geq n^{\beta}$ for some $\beta > 0$, the expression is summable and the result holds. $\qquad\square$

Combining the results of Lemmas 5.1, D.4, and D.6 gives us the main theorem, which we state here for completeness.

**Theorem** (5.3, main text). *If the tail of $\nu$ is regularly varying as per Equation* (D.3)*, then as $n \to \infty$,*

$$|V_n| \overset{a.s.}{=} \Theta(n^{\alpha}\ell(n)), \qquad |E_n| \overset{a.s.}{=} \Theta(n), \qquad |\bar{E}_n| \overset{a.s.}{=} O\left(\ell(n^{1/2})\min\left(n^{\frac{1+\alpha}{2}}, \ell(n)n^{\frac{3\alpha}{2}}\right)\right).$$

**Remark D.7.** *Finally, to conclude that there exists a class of sparse, edge-exchangeable graphs, we examine the asymptotics from this result in more detail. In the multigraph case, we see that the number of vertices increases at the same rate as $n^{\alpha}\ell(n)$, and the number of edges increases linearly in $n$. So $|E_n|$ grows at the same rate as $|V_n|^{1/\alpha}\ell(n)^{-1/\alpha}$. When $\alpha \in (1/2, 1)$, the exponent $1/\alpha$ lies in the range $(1, 2)$, and thus this parameter range for $\alpha$ results in sparse graph sequences. For binary graphs, the number of edges $|\bar{E}_n|$ grows at a rate that is bounded by $\ell(\sqrt{n})\min\left\{|V_n|^{\frac{1+\alpha}{2\alpha}}\ell(n)^{-\frac{1+\alpha}{2\alpha}}, |V_n|^{\frac{3}{2}}\ell(n)^{-\frac{1}{2}}\right\}$. Since $\min\left\{\frac{1+\alpha}{2\alpha}, \frac{3}{2}\right\} \leq 3/2 < 2$, binary graphs are sparse for any $\alpha \in (0, 1)$. Note that $\ell(n)$ does not affect the growth rate throughout since it is a slowly-varying function; i.e., for all $c > 0$, $\ell(cn) \sim \ell(n)$. For the three-parameter beta process, which we examined in our simulations, the function $\ell$ is a constant function, as in Equation* (D.5)*.*

We have shown that edge exchangeability admits sparse graphs by proving the existence of sparse graph sequences in a wide subclass of graph frequency models: those frequency models with weights generated from Poisson point processes whose rate measures have power law tails. Notably, we have shown the existence of a range of sparse and dense behavior in this wide class of graph frequency models, as desired.

## Footnotes

[1]This is conditional independence since the $(w_j)$ may be random.