[Reviews · NeurIPS 2016]

Reviewer 1

Summary

The paper considers an alternative notion of symmetry for random graphs, edge exchangeability, and studies the properties of some network models within this framework. In particular, the authors show that their class of models can produce sparse graphs, where the number of edges scales subquadratically with the number of nodes.

Qualitative Assessment

The paper is well written and the novel class of network models clearly of importance. The authors defined how edge-exchangeability differs from the classical notion of vertex exchangeability, and the connections between their class of network models and the class of models based on exchangeable random measures. The authors consider two constructions for multigraphs: (1) with single edge per step (2) with a random number of edges per step. A simple graph is then obtained by ignoring multiple edges. The authors focus on a constructions based on completely random measure and show that, for some values of the parameters, it yields sparse multigraphs and simple graphs. Overall, I like the paper, the content is technically correct and of importance. My main concern is the large overlap with the paper of Crane and Dempsey (CD), cited by the authors, line 188-190. CD basically consider the exact same notion of edge exchangeability (although in the wider framework of hypergraphs). CD focus on a particular model, the hollywood model, which belongs to the class of "single edge per step": In this case, the measure W is defined as the product measure of a Pitman-Yor random measure by itself. CD also derive sparsity properties for their hollywood process, which are similar to those obtained by the authors here. I have the impression that both works have been conducted in parallel. In a sense, the sparsity results of this paper are more general than those of CD, which have been derived only for the Pitman-Yor based model. I feel however that the "single edge per step" is much more natural than the model with a random number of edges. In particular, inference is not discussed at all in the present paper, and I suspect it is not trivial in the "multiple-edges" case. In any case, the paper should include a much more detailed description of the connections with the work of Crane and Dempsey. Minor comments: * Theorem 5.3: It would be good to mention that \ell is a slowly varying function, and the tail Levy intensity is a regularly varying function, and reference e.g. Feller. * The sparsity results are different from those obtained with the model of Caron and Fox. In the Caron-Fox case, the multigraph and simple graphs have the same growth rate, whereas here they have different rates; a discussion of this point may be useful. * I think an early contribution in the class of models discussed in this paper was made by Sinead Williamson, where W is taken to be a product of a Dirichlet random measure by itself: http://www.gatsby.ucl.ac.uk/~szabo/ml_external_seminar/Sinead_Williamson_external_seminar_20_05_2015_slides.pdf I don't think there is any referenced paper though.

Confidence in this Review

3-Expert (read the paper in detail, know the area, quite certain of my opinion)


Reviewer 2

Summary

The paper proposes a notion of edge-exchangability for random graphs and shows (theoretically and emirically) that edge-exchangable graphs can be sparse.

Qualitative Assessment

A very interesting read that I think will be of broad interest. It is primarily a theoretical work, it is very well written, but at times (due to its technical nature) a bit hard to follow, but I think overall the authors made the right decision regarding what to include in the manuscript. The comparison with the Caron-Fox approach is very good and useful for the reader. What I think could make the paper even better would be a bit more focus on how these ideas can be put to use and the implications e.g. in practical modeling of network data.

Confidence in this Review

2-Confident (read it all; understood it all reasonably well)


Reviewer 3

Summary

The authors provide a novel model for generating a family of random graphs, namely the edge-exchangeable model. The edge-exchangeable model is characterized by the invariance with respect to permutations of the order in which the edges are added to the graphs. What is really interesting about this model is that it allows the generation of a sparse family of random graphs, which is not possible with vertex-exchangeable models (recall that vertex-exchangeable models are models that are invariant with respect to a permutation of its vertices).

Qualitative Assessment

The paper is very well written, with good arguments and justifications to all claims. The definitions and results are well motivated and the simulations are well designed.

Confidence in this Review

2-Confident (read it all; understood it all reasonably well)


Reviewer 4

Summary

This paper proposes edge exchangeability in random graph sequences. Compared to the popular vertex exchangeability that leads to either dense graph or empty graph, the authors show that edge-exchangeable models can generate sparse graphs using poisson point process construction. Sparse graphs are better than dense graphs, since sparse graph sequences are more representative of real-world graph behavior.

Qualitative Assessment

I agree with the major part of this paper which shows that edge-exchangeable models can generate sparse graphs, and I also agree that sparse graph is effective in many scenarios. For example, sparse graph generated by sparse manifold clustering and embedding (in NIPS 2011) has been proved to recover the underlying manifold structure (or approximate manifold structure) of complex data. My biggest concern is that, according to the described generation process for the sparse graphs, it is not clear whether the a sparse set of edges induced by the way described in section 5 of the paper can reveal the intrinsic sparse connection between the vertices. Interpreting each data point as a vertex in the graph, the typical benefit of sparse graph comes from connecting the vertex to others that lie in the same manifold. Based on the description of the Poisson point process and edge-exchangeable models, it is difficult to see if the generated sparse graphs enjoy the mentioned typical benefit exhibited by existing sparse graph models. While I agree with the importance of sparsity in the graphs, pursuing sparsity solely for the purpose of sparsity may not lead to a sparse graph with good performance in practice.

Confidence in this Review

1-Less confident (might not have understood significant parts)


Reviewer 5

Summary

This manuscript tries to provide a new measure to model the large sparse networks, which is different from Caron’s Kallenberg exchangeability. However, the exchangeability and sparseness can not be kept at the same time in the model. After pre-processing on the edges’ counting (which leads to nonexchangeable), the author has shown the edge’s number is subquadratic to the vertex’s number.

Qualitative Assessment

When comparing to Caron's Kallenberg exchangeability, the author claims the non-stationary domain issue. However, as far as I can tell, this stationary domain of the proposed model does not guarantee better results. Can you please show some reasons for constructing stationary domain? The proposed graph-frequency model works as if the de-finetti theorem. My main question over this is whether the spatial structure of network would be lost? Especially, the location parameter \theta seems always to be unimportant in constructing the network. If without this location parameter, can we still get some insight into the network structure? Another issue lies in Line 162-165 in page 4. As the author has admitted, the model needs to processed to be sparse (under some specific Poisson process condition), while at the same time the exchangeability can not be kept. In this sense, exchangeability and sparse modeling can not be modeled at the same time. The author should make clear on this point in both the abstract and introduction. The author also should discuss more on Crane's edge-exchangeability and the Hollywood process. From my understanding, Crane's Hollywood process works as if the CRP, while the measure proposed by this manuscript works as if the stick-breaking process for CRP. However, this measure does not work as well as the Hollywood process, which is a bit disappointment. As a result, the potential impact is reduced.

Confidence in this Review

2-Confident (read it all; understood it all reasonably well)


Reviewer 6

Summary

This paper introduces an alternative notion of exchangeability for random graphs, called edge exchangeability, by viewing a graph as a sequence of edges. In contrast to the classical Aldous-Hoover representation where the number of edges increases in proportional to the square of the number of vertices (a.s. dense or empty), the proposed graph representation is not only exchangeable in terms of edges but also leads to sparsity, i.e., the number of edges is asymptotically upper bounded by constant*#_vertex^2. The authors also propose the graph frequency model and simulate its generative process to validate the sparsity property.

Qualitative Assessment

[Technical quality] In Section 2.2, the authors have elaborated the step-augmented graph representation for edge-exchangeable graph sequences and try to define the edge exchangeability on the permutation of steps. But to me it still looks like directly applying the de Finetti's theorem to an exchangeable sequence conditioned on W Eq.(1), where the elements are pairs of vertices. Although W is two-dimensional and \theta has two indices, it actually is treated as a sequence (1-array), so the de Finetti's theorem can be directly applied. Is my understanding right? Or it is indeed a new notion of exchangeability? The biggest flaw of this work may be that: According to the measure defined in Eq.(1), the generative process only generates a multi-set, whose elements are i.i.d. generated edges given W. However, a graph, if represented in edges, is a binary set. In Line 164, an additional operation is required to obtain the final graph by binarizing the multigraph. Thus the resulting binary graph is no longer edge-exchangeable and the measure define in Eq.(1) is not applicable to random binary graphs. Can the rate w_{I,j} only be defined in the form of w_i w_j in Eq.(3)? All the discussions in the reminder of the paper are based on this definition. Any other form? First mentioned in the introduction (Line 46) and some places later, the notion of “stationary” is not described or defined. Is it the same meaning defined in Section 7.8 [19]? And this “stationary” property is not further discussed or proved in the rest of the paper. [Novelty] On one hand, the proposed edge-exchangeability is an application of the de Finetti's theorem to an exchangeable sequence conditioned on W, which is not novel. On the other hand, the proposed graph frequency model tries to grow the number of vertices in the graph through increasing the number of steps; and tries to make connection to the well known Bayesian nonparametric model such as CRP by viewing Eq.(1) as DP and the posterior graph after n steps as CRP. Such interpretation and connection may be interesting. [Impact] Graph representation for sparse graphs are very important and can be impactful. [Presentation] The presentation and clarity of the paper is good. The running examples and connection to other forms of exchangeability in the supplementary material are helpful to understand the idea. Example 4.1 is not clearly described, and in Line 192, should it be Section 3?

Confidence in this Review

2-Confident (read it all; understood it all reasonably well)